# Phenotypic and Genotypic Diversity of Roots Response to Salt in Durum Wheat Seedlings

**DOI:** 10.3390/plants12020412

**Published:** 2023-01-16

**Authors:** Ieva Urbanavičiūtė, Luca Bonfiglioli, Mario A. Pagnotta

**Affiliations:** Department of Agricultural and Forest Sciences, Tuscia University, Via S. C. de Lellis, 01100 Viterbo, Italy

**Keywords:** durum wheat, salt stress, roots, high-throughput phenotyping, genotyping, SSR, QTL

## Abstract

Soil salinity is a serious threat to food production now and in the near future. In this study, the root system of six durum wheat genotypes, including one highly salt-tolerant (J. Khetifa) used as a check genotype, was evaluated, by a high-throughput phenotyping system, under control and salt conditions at the seedling stage. Genotyping was performed using 11 SSR markers closely linked with genome regions associated with root traits. Based on phenotypic cluster analysis, genotypes were grouped differently under control and salt conditions. Under control conditions, genotypes were clustered mainly due to a root angle, while under salt stress, genotypes were grouped according to their capacity to maintain higher roots length, volume, and surface area, as J. Khetifa, Sebatel, and Azeghar. SSR analysis identified a total of 42 alleles, with an average of about three alleles per marker. Moreover, quite a high number of Private alleles in total, 18 were obtained. The UPGMA phenogram of the Nei (1972) genetic distance clusters for 11 SSR markers and all phenotypic data under control conditions discriminate genotypes almost into the same groups. The study revealed as the combination of high-throughput systems for phenotyping with SSR markers for genotyping it’s a useful tool to provide important data for the selection of suitable parental lines for salt-tolerance breeding. Nevertheless, the narrow root angle, which is an important trait in drought tolerance, is not a good indicator of salt tolerance. Instated for salt tolerance is more important the amount of roots.

## 1. Introduction

According to the FAO 2021 global map of salt-affected soils [1], around 400 million hectares of topsoil and more than 800 million hectares of subsoil are affected by salt. Moreover, FAO reported that soil salinity per year reduces production potential by taking up to 1.5 million ha of farmland out of production and more than 30 million USD as an annual loss in agricultural productivity [1]. Unfortunately, at such rates, 50% of total cultivated land is expected to be salinized by 2050 [2] as a consequence of the phenomenon of “desertification,” and land degradation due to climate change and human action by poor land management [3,4,5]. Projections up to 2099 assume an extreme increase in drought and the subsequent soil salt accumulation in Eurasia, Africa, and Australia [6]. Soil salinization may occur through a natural process (primary salinization) or by human management (secondary salinization), through irrigation with salty water, excessive use of fertilizers, and intensive agriculture practices [7,8,9].

Salt, mainly NaCl, in the soil slows down plant development due to two stresses. At first, the plant experiences a physiological drought (osmotic stress), which reduces leaves and root growth due to the inhibition of cell expansion, cell wall synthesis, and stomata conductance [10,11]. Later the plant response is related to the toxic effect of salt, where plants accumulate toxic Na^+^ and Cl^−^ in old leaves over a long period. Due to the harmful effects of salt on germination, water uptake, transpiration, photosynthesis, enzyme activities, and metabolism of protein and lipids, a significant reduction in yield of up to 40% in agricultural lands has been reported [12,13]. Since the three most widely consumed kinds of cereal (maize, rice, and wheat) are susceptible to saline soil conditions [14], it is essential to improve studies about mechanisms to induce salt stress tolerance to secure our future on food storage.

Plants could tolerate salt stress by several strategies: (i) salt exclusion, retaining the Na^+^ in woody stems or roots; (ii) regulating toxic ions balance through the maintenance of a high K^+^/Na^+^ ratio with a mechanism of translocation of K^+^ over Na^+^, to reduce the Na^+^ concentration in the shoot cytoplasm; and (iii) by salt compartmentalization in vacuoles, protecting the cytoplasm from the ion toxicity [15,16]. The root system under an excessive salt concentration in the soil due to toxic ions accumulation suffers from inhibition of nutrition supply and elements adsorption [17]. Salinity, as drought, hurts the whole root system due to reduced cell activity in the root meristem [18]; however, growth rate reduction was determined more in the main root than in lateral roots [19]. As a response to stress, plants can remodel root system architecture (RSA) to adapt to adverse conditions [20]. However, the direction of root growth as a response to external stimuli differs in salt and drought conditions. Under drought conditions, roots use positive hydrotropism, growing towards the water, but in the presence of salt, roots remodeling due to negative halotropism directs the growth away from salinity. Moreover, it was reported that the roots of Arabidopsis exhibited reduced gravitropism under salt conditions to grow against the gravity vector [21]. In addition, the identified gene in cereal crops responsible for gravitropism versus anti-gravitropism regulatory mechanisms also controls the root angle [22]. The narrow root angle was determined as a very important feature in selecting genotypes for drought tolerance since genotypes with a narrow root angle can grow deeper compared with varieties with a wider root angle [23,24,25]. However, root system remodeling strategies to tolerate drought and salt stress differentiate in terms of tropism type, and it does not mean that a narrow root system able to adapt to drought is also capable of growing in saline soil.

In traditional breeding programs, phenotyping the above-ground plant parts was the main aspect of determining resistance to abiotic stress. However, the development of several high-throughput methods and techniques for root phenotyping in recent years has finally made it possible to determine which root systems and remodeling strategies are associated with the most successful plant development under stress conditions [26,27,28,29]. Phenotypic root traits could be used to determine genetic diversity; however, they are often influenced by environmental conditions. According to Soriano et al. [30], the stable quantitative trait loci (QTL) and closely linked molecular markers can improve durum wheat genetically through marker-assisted selection, cloning, and pyramiding in breeding programs to create new cultivars with desired quantitative traits. Molecular markers such as simple sequence repeat (SSR) and its based association mapping analysis may allow us to determine the chromosomal regions involved in root architecture features [30,31,32,33]. Moreover, SSRs are chromosome-specific and highly effective molecular tools due to their large level of polymorphism, distributed over all of the genomes [34,35]. Simple sequence repeat (SSR) molecular markers closely linked with genome regions associated with root architecture features could be helpful in selecting genotypes for salt tolerance breeding programs. Moreover, the identification of closely linked candidate genes can give a better understanding of the genetic response to specific stimuli.

In this study, six durum wheat genotypes, including one high salt-tolerant J. Kethifa as check genotype, were used to evaluate differences in shoot and root traits under control and salt conditions at the seedling stage and to determine their involvement in salt tolerance. The root system was evaluated using a high-throughput phenotyping system. The genotyping was performed using 11 SSR markers closely linked with genome regions associated with root traits, and in silico, candidate gene expression analysis was performed. The study aimed to evaluate genetic diversity among genotypes based on phenotypic and genotypic variations to identify candidate parental genotypes for salt-tolerance breeding.

## 2. Results

### 2.1. Salt Effect on Durum Wheat Shoots Traits

Highly statistically significant differences were detected between genotypes and both treatments (salt vs. control) for seedling length (PH), the number of leaves (NL), and the number of tillers (NT) (Table 1). Moreover, significant genotype by treatment interaction highlighted that the different genotypes used different strategies in terms of shoot development under salt stress. For example, the genotypes Azeghar and Sebatel had not shown a significant salt effect on any shoot traits and maintained one tall main stem with 5–7 leaves. In the genotypes, Cham1 and Pelsodur seedling length and the number of leaves were not significantly affected by salt; however, there was a significant negative reduction in the number of tillers. The salt-tolerant genotype J. Khetifa as a check plant, under salt stress, reduced seedling length and significantly increased the number of leaves (Table 1).

### 2.2. Salt Effect on the Root System

The analysis of variance revealed the presence of significant differences between genotypes and treatments for all root traits except average root diameter (AG) (Table 2). Significant interaction effects were detected for root angle (RA) and the number of tips (TI).

Azeghar, which maintained a wide root angle under both conditions, had a significant positive effect of salt only on root volume and surface area (Table 3). While Cham1, with wide and not affected by salt root angle as Azeghar, had a significant positive effect on root length (RL) that increased significantly by more than 80%, and the number of tips (TI) by more than double, and negative effect on root mean diameter (Table 3). Sebatel, Vulci, and Pelsodur showed a significant effect of salt only on root angle, which expanded in the presence of salt in the soil; this was particularly evident for Pelsodur, where the differences were around 40%. The highest positive significant effect on the root system under salt stress was determined for the salt-tolerant genotype J. Khetifa. Features such as root length (RL), root volume (RV), root surface area (SA), and the number of tips, forks, and crossings doubled under salt stress in seedlings of J. Khetifa. Under control conditions, Azeghar had a higher root length (RL) than J. Khetifa and Cham1, while under salt stress J. Khetifa maintained the longest roots, Sebatel and Azeghar had medium root length, and Vulci, Pelsodur, and Cham1 had the shortest. Under control conditions, Azeghar, Sebatel, and Cham1 had a wide root angle (RA), while J. Khetifa, Pelsodur, and Vulci had a narrow RA. However, RA widened significantly only for Pelsodur, Sebatel, and Vulci under stress. Under salt conditions, the mean root diameter (AG) significantly decreased only for Cham1 by about 24%.

### 2.3. Correlation among Traits

The correlation matrix among traits (Figure 1) shows that the root angle under both conditions (RAC and RAS) had a strong significant (*p* < 0.05) negative correlation with shoot traits such as the number of leaves and tillers under control (NLC, NTC) and salt (NLS and NTS) conditions. It could be suggested that genotypes with steeper root angles tend to produce more tillers and leaves.

Moreover, root angle under both conditions (RAC and RAS) had a highly significant positive correlation with root volume and surface area under control conditions. An interesting association was found between the root length (RL) and root traits that show the formation of new roots, such as the number of tips (TI), forks (FR), and crossings (CR). Root length under control (RLC) had a significant (*p* < 0.05) positive correlation with these traits (TIC, FRC, and CRC) only under control conditions and RLS with the same traits only under salt conditions. Moreover, root length under salt conditions (RLS) had a strong positive correlation with root volume (RVS) and root surface area (SAS) under salt conditions. In general, a significant positive correlation was found between root volume (RV) and surface area (SA) under both conditions, while both of them had a positive correlation with root angle (RAC) only under control conditions. 

### 2.4. Principal Components Analysis (PCA)

The top two principal components of PCA account for 82.6% of the total phenotypic variation under control conditions (Figure 2A). PC1 explained 49.1% and was mainly associated with root traits such as -RL, -SA, -TI, -FR, -CR (Appendix A). At the same time, PC2 explained 33.5% of the total variation and was strongly associated with shoot trait and root angle (Figure 2A). Genotype J. Khetifa was grouped with Vulci and Pelsodur on the positive side, while Azeghar, Sebatel, and Cham1 were on the negative side of PC2. However, J. Khetifa with Cham1 was assigned on the PC1 positive side, Azeghar and Sebatel on the negative, and Vulci with Pelsodur were close to zero value. Three clusters of genotypes were identified based on hierarchical classification under control conditions (Figure 2B). One cluster grouped J. Khetifa, Vulci, and Pelsodur, the second group Sebatel and Azeghar; while Cham1 separated from the rest genotypes quite far as a third group. Under salt conditions, the top two principal components of PCA account for 92.3% of the total variation of root and shoot traits (Figure 3A). PC1 explained 57.5% and divided genotypes Vulci, Pelsodur, and Cham1 on the positive side of PC1, while J. Khetifa, Azeghar, and Sebatel were on the negative side.

PC2 explained 34.8% and was associated with shoot trait and root angle the same as under control conditions (Figure 3A). J. Khetifa with Vulci was assigned on the PC2 positive side, Azeghar and Sebatel on the negative, and Cham1 with Pelsodur were close to zero value. According to hierarchical analysis, all genotypes were distinguished into two major clusters, where Sebatel and Azeghar were assigned together with the salt-tolerant J. Khetifa to one cluster, and Vulci, Pelsodur, Cham1 were grouped to the second one (Figure 3B).

### 2.5. Genetic Diversity Based on SSR Markers

Eleven co-dominant SSR markers associated with specific root morphological traits were used to determine the genetic characteristics of the six genotypes. The number of alleles detected for each marker was very variable, ranging from 2 for wms205 to 5 for cfa2257, with an average of about three alleles per marker (Table 4). Some markers, i.e., gwm234 and wmc727, identified two loci labeled with a and b after the SSR name.

The observed heterozygosity was zero for most of the markers. Nevertheless, despite working with a self-pollinated species, the observed heterozygosity was not always equal to zero. Some markers detected high observed heterozygosity (i.e., wmc727a, wms205, and gwm573.2). Regarding the heterozygosity of wms205, it should be noted that all six genotypes have the same allelic situation leaving a suspect of a non-perfect allelic situation among the two detected bands common for all the genotypes. The gene diversity, computed as expected heterozygosity (He), and the polymorphism information content (PIC) provides information on a marker’s ability to determine polymorphism. In the present study, the gene diversity is quite high, ranging from 0.278 (gwm234a and wmc727b, which are second loci detected by an SSR marker) to 0.778 (cfa2086) (Table 4). The PIC values had a similar but not equal ranking among markers compared with the gene diversity parameter. PIC ranged from 0.24 for gwm234a and wmc727b to 0.74 for cfa2086 (Table 4). The Shannon Information Index (Table 4) indicates richness, and the evenness is always quite high ranging from 0.45 to 1.56, with an average of 1.01. The polymorphism among the different genotypes was not dramatically different; it ranged from about 13% in Pelsodur to about 27% in Azeghar and Sebatel, with an average of 21% (data not shown). Interestingly the number of Private alleles, i.e., alleles present only for a single genotype, is quite high (Table 5). All the genotypes have at least one allele present only in those genotypes. This goes from Vulci with one private allele to Azeghar and Pelsodur with four private alleles. Not surprisingly, cfa2086, with the higher PIC, also has a higher number of private alleles detected.

The UPGMA phenogram of the Nei (1972) [36] genetic distance cluster for 11 SSR markers (Figure 4) and all phenotypic data under control conditions (Figure 2B) discriminate the tested genotypes almost into similar clusters. Genotypes with wide root angles, such as Azeghar and Sebatel, were grouped morphologically and in terms of genetic distance. Another cluster, according to genetic distance, consisted of genotypes with narrow root angles, such as J. Khetifa, Pelsodur, and Vulci (Figure 4). Cham1, mainly in terms of phenotypic, was far from other genotypes, while genetically is far from the other genotypes but closer to Azeghar and Sebatel.

### 2.6. More Detailed Analysis of Genomic Regions Linked to Selected SSRs Markers and In Silico Candidate Gene Expression Analysis

The SSRs markers used in this study are distributed in seven chromosomes and overlap with 32 root-related QTLs (Appendix A). For example, wms5, located on chromosome 3A in the durum wheat genome, overlapped with seven root-related QTLs for the number of root tips, primary root diameter, primary root length, lateral root diameter, total lateral root surface and volume, and total root number. Markers gwm427 and gwm459 located on chromosome 6A overlapped in total with eleven QTLs associated with root traits such as total root length, root angle, average root length, primary root length, root surface area, root tips, and total lateral root surface and volume, primary root volume. Markers gwm499 and gwm234, located on chromosome 5B, overlapped with three QTLs related to total root length, total lateral root length, and primary root volume. Gwm636 and cfa2086 markers on chromosome 2A overlapped with QTLs related to lateral root number per primary root, total root number, and root growth angle. Markers wmc727 and wms205 markers located on chromosome 5A overlapped with QTLs related to average root length and primary root length. Cfa2257 marker located on chromosome 7A overlapped with Meta QTL associated with total root number and primary root length. Marker, gwm573.2, located on chromosome 7B, overlapped with root traits related to Meta QTL.

Candidate genes for investigating and estimating gene expression levels were identified according to SSRs location on the durum wheat ‘Svevo’ genome (Appendix A). A total of 57 gene models were detected and analyzed at http://www.wheat-expression.com/ (accessed on 13 January 2023), using their RNAseq to obtain the expression of genes involved in root response to abiotic stress at the seedling stage (Appendix A).

In the durum wheat genome, (RefSeq v1.0 Chr 2A region), the mapped marker gwm636 was close to nine genes models associated with growth and tolerance to abiotic and biotic stresses, such as cell differentiation protein RCD1, sterol 3-beta-glucosyltransferase, cytochrome P450, Sec24-like transport protein, AAR2 family protein, leucine-rich repeat receptor-like protein kinases. All genes were up-regulated in seedlings’ roots under abiotic stress (Appendix A). The marker wms5 was close to seven genes models associated with response to stress or plant development under stress conditions, including thioredoxin-like protein, homeobox leucine-zipper protein, citrate-binding protein, DUF1677 domain-containing protein, AGAMOUS-like MADS-box protein, kinase superfamily protein, and core-2/I-branching beta-1, 6-N-acetylglucosaminyltransferase family protein. Five of these were up-regulated in seedlings’ roots under abiotic stress. Wmc727 on chromosome 5A was linked to 10 genes models mostly involved in the regulation of secondary metabolites related to antioxidant activity, metabolic pathways, transcription, and signaling, such as membrane protein insertase YidC, O-methyltransferase family protein, nuclease S1, short-chain dehydrogenase/reductase family protein, root phototropism protein, Chalcone, and Curcuminoid synthases, F-box-like protein, and Palmitoyl protein thioesterase containing protein. Eight from 10 genes linked to marker gwm499 on chromosome 5B were close to genes associated with response to abiotic stress, such as receptor kinase, basic helix-loop-helix transcription factor, and ADP-ribosylation factor GTPase-activating protein involved in the root growth direction, and was most expressed gene near gwm499 marker. The marker gwm427 on chromosome 6A was linked to seven genes models related to plant’s adaptation and tolerance to abiotic stress, such as actin-related protein 2/3 complex subunit, plastid movement impaired 1-related 1 G protein, cytochrome P450, inosine-5’-monophosphate dehydrogenase, CBS domain-containing protein. All genes linked to marker gwm427 were expressed under abiotic stress in seedling roots. The marker gwm573.2 was close to 21 gene models associated with plant growth and development under abiotic stress such as trehalose 6-phosphate phosphatase, a protein of the DNA-directed RNA polymerase the ABC transporter, phosphatidylinositol: ceramide inositol phosphotransferase, shikimate kinase, G-protein, ATP-dependent zinc metalloprotease FtsH1, Aldo/keto reductase family oxidoreductase, Sec14p-like phosphatidylinositol transfer family protein, Pentetratricopeptide repeat proteins, a bHLH transcription factor, the regulator of chromosome condensation (RCC1) family with FYVE zinc finger domain-containing protein, HVA22-like protein, TCP transcription factor, Photosystem (PS) II CP43 reaction center protein, BTB/POZ domain-containing protein, Sphingoid base hydroxylase 2, and two F-box-like protein. Almost all of them were up-regulated in wheat seedlings’ roots under abiotic stress conditions (Appendix A).

## 3. Discussion

The root system remodeling strategies under adverse conditions is a great opportunity for adaptation and optimal development of the plant, as the first organ that senses and responds to abiotic stress, and then through efficient uptake of water and nutrients. Latest studies demonstrated that root system features, especially root angle in crop plants, play a main role in adapting to adverse conditions [20,22,37].

According to Pariyar et al. [38], together with root traits, leaf-related traits should be included for more efficient selection in breeding programs for whole plant establishment. At the same time, the ideotype of the root system plays a key role in ensuring sufficient leaf area, which in turn is responsible for efficient photosynthesis, the crucial process for plant development. Genotypes that produce new leaves faster than the die of old ones and maintain a high number of leaves or shoot biomass under stress conditions could be considered tolerant. Moreover, previous studies have shown that salt-tolerant genotypes of durum wheat accumulated higher shoot biomass and maintained a bigger leaf area under salt stress compared with susceptible varieties [39,40,41]. In our study, some shoot traits were associated with root angle. For example, the salt-tolerant genotype, J. Khetifa, under salt stress, maintained a narrower root angle and the highest number of leaves. Moreover, Vulci, genetically close to J. Khetifa, was characterized by a narrow root angle and a large number of leaves under salt conditions. However, in our previous drought study [25], Pelsodur, under control and drought conditions, maintained a narrow root angle, which was associated with good shoot development and drought tolerance. While in this study, under salt stress, the Pelsodur root angle changed significantly from narrow to wide, and this caused a significant negative effect on his shoot development, especially tillers that decreased by 60%. Moreover, in genotypes with wider root angles, such as Sebatel, Cham1, and Azeghar, the number of leaves was not significantly affected by salinity. It is well known that a narrow root angle is one of the main traits for drought tolerance, but it was demonstrated that a shallower root growth angle could enhance rice yields in saline environments [20].

Phenotypic data from our pot experiment obtained using the WinRHIZO software (version 4.0b) highlighted a few trends in plant response to salt stress at the seedling stage. Genotypes under control conditions were distinguished regarding root angle into two groups, genotypes Vulci and Pelsodur with high salt-tolerant J. Khetifa had a narrower root angle than genotypes Sebatel, Azeghar, and Cham1. It was reported that root angle depends on plant root ability to control gravitropic versus anti-gravitropic mechanisms, which is the main factor in avoiding salty soil [20,21,22]. In this regard, root tips could be a good indicator for selection as they are involved in root gravity perception and regulation. In addition, the salt-tolerant genotype J. Khetifa under salt stress doubled the number of root tips and had a significantly higher number compared with the other genotypes. However, under salt conditions, the salt-tolerant J. Khetifa, with a narrow root angle, was grouped with genotypes Azeghar and Sebatel, which had a wider root angle. These results suggest that a narrow root angle is not a good indicator of salt stress tolerance.

In general, the check genotype J. Khetifa under salt stress at the seedling stage improved the number of leaves, which is a good salt tolerance indicator in terms of accumulation of toxic ions and efficient photosynthesis. Moreover, J. Khetifa improved almost all root traits under salt conditions, except the root angle, which stayed narrow, and the mean root diameter. The other two genotypes, Azeghar and Sebatel, from the same cluster of J. Khetifa under salt stress, had wide root angles, and salt did not affect shoot traits significantly. Moreover, genotype Azeghar under salt increased root volume and surface area, and the roots of Sebatel were not affected significantly. Based on present results, it can be concluded that a narrow root angle is not a good indicator of salt tolerance, moreover, exist more root system adaptation strategies that depend on the inner capacity of genotypes to maintain an efficient amount of roots.

Dendrograms based on the cluster analysis of the genetic similarity coefficients and phenotypic data under control conditions grouped genotypes mainly due to root angle. In this regard, the SSR marker wms5 can be quite informative in selecting genotypes for narrow-angle root system ideotype. It was reported that the wms5 marker was significantly associated with root angle [42], and it was located on chromosome 3A in the durum wheat genome. In our study, genotypes grouped in the same clusters according to root angle, such as J. Khetifa, Vulci, and Pelsodur, had narrow root angles (wms5 band of 190 bp), and Sebatel and Azeghar (wms5 band of 180 bp), Cham1 (wms5 band of 185 bp) had wide root angles. Moreover, in the durum wheat genome, the location of wms5 was close to gene models representing proteins involved in plant development and response to stress. Such as thioredoxin, which is a small ubiquitous protein that reduces protein disulfides, protects cells from oxidative damage, and protects plants from the oxidative deactivation of photosynthesis-related enzymes [43]. Moreover, homeobox leucine-zipper transcription factors are responsible for plant growth adaptation in water deficit conditions [44,45]. Core-2/I-branching beta-1,6-N-acetylglucosaminyltransferase family protein is involved in cell elongation during the seedling growth stage and senescence of the leaf process [46]. Citrate-binding protein family is involved in plant development and growth, also in response to stress related to the accumulation of toxic ions in the vacuoles [47]. DUF1677 family proteins are DUF domain-containing proteins that play a vital role in plant stress responses, including salinity [48,49]. Kinases superfamily proteins are well known as signal transmitters in various stresses [2]. Members of the MADS-box gene family are involved in many processes of plant development and morphogenesis, but it was also reported to modulate root architecture [50,51].

Our results revealed that the root system under the salt of genotypes Azeghar and Sebatel were close to the one of salt-tolerant J. Khetifa, regardless of root angle differences, making them good candidates as parent lines in breeding programs. In this respect, some rare alleles of SSR markers could help to select salt-tolerant candidates in the segregant populations. For example, the salt-tolerant J. Khetifa and Sebatel had rare alleles near marker cfa2086, 277 bp, and 223 bp, respectively. Moreover, salt-tolerant J. Khetifa had rare alleles by SSR marker gwm234 (214 bp) and near marker gwm499 (154 bp). Moreover, genotype Sebatel had a rare allele also near marker gwm499. In the durum wheat genome, the mapped marker gwm499 was close to gene models for kinases superfamily protein, which mostly regulates transcription factors. Moreover, it was near to basic helix-loop-helix transcription factors, which are involved in a range of abiotic stress responses and adaptive responses to drought [52]. ADP-ribosylation factor GTPase-activating protein was shown that is crucial for determining directional root hair growth in Arabidopsis [53].

Genotypes Azeghar and Sebatel had rare alleles near marker gwm427, 201 bp, and 237 bp, respectively. The marker gwm427 position was found to overlap with previously reported durum wheat major QTL for root growth angle (qSRA-6A), where candidate gene analysis showed loci related to gravitropism, polar growth, and hormonal signaling [54]. In the durum wheat genome (RefSeq v1.0 Chr 6A region), the mapped marker gwm427 was close to genes models for actin-related protein and plastid movement-impaired related protein. Marker gwm427’s location in the durum wheat genome was close to two related genes such as (i) the actin-related protein and (ii) the plastid movement impaired 1-related 1 G protein that regulates organelle movement [55]. In general, the movement and positioning of the organelles play an important role in the adaptation of plants to environmental stress. Moreover, actin-related protein regulates developmental processes such as cell division and development and cell polarity [56]. Marker gwm427 location in the durum wheat genome was close to other five gene annotations (Appendix A); (iii) cytochrome P450 from the oxidoreductases enzymes class has been reported to be involved in plant defense against various abiotic and biotic stresses [57]. (iv) Inosine-5’-monophosphate dehydrogenase catalyzes the biosynthesis of the guanine nucleotide, which is important for DNA and RNA synthesis, signal transduction, energy transfer, glycoprotein synthesis, and cellular proliferation [58]. (v) Proline-tRNA ligase is an enzyme involved in proline metabolism and KEGG and aminoacyl-tRNA biosynthesis pathways (BRENDA:EC 6.1.1.15). L-Proline in a stressful environment is crucial for plants due to its protective role as an osmoprotectant and stabilizes cellular structures and enzymes, and scavenging reactive oxygen species [59]. (vi) Potassium-transporting ATPase could help plant cells to maintain a high concentration of potassium and a low concentration of sodium inside in case of high sodium concentrations in the soil. (vii) Gene containing cystathionine β-synthase (CBS) domain was found to be upregulated under high salinity, and overexpressing this gene showed higher abiotic stress tolerance in transgenic plants [60].

Genotype Azeghar had three more rare alleles, two near marker cfa2257 (104 bp and 142 bp) and one near gwm573.2 (181 bp) (Appendix A). The marker gwm573.2 was close to gene models translated to trehalose 6-phosphate phosphatase, which regulates the activity of protein kinase, and this has a large impact on cell division, development, and function [61]. The protein of the DNA-directed RNA polymerase belongs to complex molecular machines for the essential function of the synthesis of RNA from DNA templates, otherwise transcription [62]. Abscisic acid is a substrate of the ABC transporter that is involved in the active transport of various molecules across biological membranes, such as heavy metals, lipids, glucosinolates, and phytohormones [63] allowing to coordinate of physiological and developmental processes at the whole-plant level and being involved in response to several induced by biotic stresses. Other related genes (Appendix A) are (i) Phosphatidylinositol: ceramide inositol phosphotransferase, which is essential for sphingolipid biosynthesis and plays an important role in dehydration stress tolerance [64]. (ii) Shikimate kinase catalyzes the reaction of the shikimate pathway that redirects carbon from the central metabolism pool to a wide range of secondary metabolites responsible for plant development, growth, and stress responses [65]. (iii) G-proteins are universal signal transducers involved in many cellular responses of plants [66]. (iv) ATP-dependent zinc metalloprotease FtsH 1 is involved in the formation of thylakoid membranes during early chloroplast development and in the proteolysis of damaged photosystem II (PSII) protein complex repair cycle, which is essential to avoid photoinhibition due to the accumulation of photodamaged PSII [67]. (v) Aldo/keto reductase family oxidoreductase is involved in many plant metabolites reactions, such as the biosynthesis of osmolytes, reactive aldehyde detoxification, secondary metabolism, and membrane transport [68]. (vi) Sec14p-like phosphatidylinositol transfer family protein is a plasma membrane-associated protein that has a crucial role in rooting patterning due to auxin signaling and PIN distribution in Arabidopsis [69]. (vii) Pentatricopeptide repeat proteins regulate chloroplast gene expression and RNA metabolism in higher plants [70]. (viii) Transcription factor bHLH involved in response and tolerance to various plant stresses, including soil salinity [71]. (ix) The regulator of chromosome condensation (RCC1) family with FYVE zinc finger domain-containing protein is essential for regulating auxin flows toward the direction of gravity and root branch angle in Arabidopsis [72]. (x) Transcriptomic analysis in roots revealed that the ABA-related gene encoding HVA22-like proteins was up-regulated, and considering that ABA biosynthesis is correlated with auxin transport to the root tips, their co-expression can be beneficial for the development and morphology of the roots [73]. (xi) TCP transcription factor may activate OsNHX1 gene expression, which response to salt and PEG-induced drought stress in rice, and may be associated with abiotic stress tolerance [74]. (xii) Photosystem (PS) II CP43 reaction center protein is the main antenna of the PS II pigment-protein complex in the thylakoid membrane and plays an essential role in the photosynthetic responses to stresses in higher plants [75]. (xiii) BTB/POZ domain-containing proteins are associated with plant growth and development, also involved in plant responses and defense against biotic and abiotic stresses [76]. (xiv) The enzyme Sphingoid base hydroxylase II is essential for growth and viability in Arabidopsis [77]. The marker gwm573.2 was also close to (xv) two F-box-like protein genes models, which were described earlier as genes involved in salt tolerance, and rooting abilities.

## 4. Materials and Methods

### 4.1. Plant Material

For phenotyping and genotyping was used six *Triticum turgidum* L. subsp. *durum* (Desf.) Husn. accessions, including one high salt-tolerant genotype Jennah Khetifa as check [78,79]. Other genotypes characterized by different root angles, Cham1, Azeghar 2-1 (56) (Azeghar), and Sebatel2 (45) (Sebatel) with wide root angle, and two genotypes Mv-Pelsodur (Pelsodur) and Vulci with narrow root angle [25].

### 4.2. Pot Experiment Condition and Design

A pot experiment was performed in a greenhouse at Tuscia University’s experimental farm (Viterbo, Italy) from January to February 2021. A Randomized Complete Block Design (RCBD) with three replications, six genotypes, and two salinity levels was used. The six genotypes were sown with one seed per pot placed with the embryo facing down on 20 January 2021. Each pot was 17 cm (diameter) by 16 cm (high) and filled with 2.5 L of sand (Appendix A). The temperature in the greenhouse was constantly maintained at around 25 °C during the day and around 15 °C during the night to have optimum growing conditions. Before starting the treatments, all pots (control and salinity treatment) were irrigated, using a dropping pipe, three times per week (80 mL/per pot) with a water nutrition solution composed of dihydrogen phosphate potassium (0.13 g/L), potassium sulfate (0.04 g/L), nitric acid (0.286 mL/L), calcium nitrate (0.432 g/L), potassium nitrate (0.436 g/L), magnesium nitrate (0.244 g/L) and Micron (0.23 g/L) [25]. Two weeks after sowing, when all seedlings reached two fully expanded leaves stage, plants under salinity treatment were started treated with a salt solution (250 mM NaCl) 3 times per week (80 mL/per pot) for 14 days, while the ones under control treatment were irrigated with the same solution but without NaCl. After two weeks of salt treatment, pots were watered with a water nutrition solution without salt for 14 days. After two weeks (no NaCl), for a week, the salt solution was applied again, and after that, the plants were removed for analysis (Figure 5).

### 4.3. Roots Phenotyping

Plants for analysis were collected after 7 weeks after sowing in total. Roots were carefully washed to remove the sand, using a soft spray watering head, and analyzed using Win-RHIZO Pro software v2009 (version 4.0b; Regent Instruments, Montreal, QC, Canada). The morphological traits of the roots were recorded for the whole root system (Figure 6A), such as total root length (RL), the sum of all roots lengths, cm, surface area (SA), total root surface area, cm^2^, root volume (RV), total root volume, cm^3^, number of tips (TI), forks (FR), and crossings (CR). The root angle (RA, degree on the vertical °) between the two extreme sides of the roots with the center set in the middle of the crown was measured using the software ImageJ, which could be freely download at https://imagej.nih.gov/ij/download.html; accessed on 13 January 2023 (Figure 6B). Morphological traits for shoots, such as plant height, number of leaves, and number of tillers, were also recorded.

### 4.4. DNA Extraction, PCR Amplifications, and Gel Electrophoresis

For genotypic diversity analysis, 11 microsatellites associated or linked to genomic regions related to root traits from previously reported studies (Appendix A) were chosen. The genomic DNA was extracted from fresh leaves of each genotype using the extraction kit PureLink Plant Total DNA Purification kit (Invitrogen; Thermo Fisher Scientific, Waltham, Massachusetts USA). The PCR amplifications were carried out using 0.125 μL of GoTaq G2 DNA Polymerase (Promega, Madison, Wisconsin, USA), 5 μL of 5X Colorless GoTaq^®^ Reaction Buffer, 0.5 μL of 10 µM dNTP mix, 2 μL of 10 µM/μL forward and reverse primer, 100 ng of gDNA, and sterile water for a total reaction amount of 25 μL. The amplifications were performed in Swift Maxi Thermal Cyclers (Esco Technologies, St. Louis, Missouri, USA) with an initial denaturation of 5 min at 94 °C followed by 35 cycles with 15 s at 94 °C and 20 s at different temperatures, according to the annealing temperature of each marker (see Table 6), and 20 s at 72 °C. As a final step, it was performed an extension of 10 min at 72 °C. The electrophoresis was carried out through the capillary electrophoresis device QIAxcel Advanced Instrument (Qiagen, Hilden, Germania) using the QIAxcel DNA High-Resolution Kit, and allele analyses were performed with QIAxcel ScreenGel Software (Qiagen, Hilden, Germania). Each allele was named for its band size in the base pair (Appendix A).

Gene models were identified in the NCBI database (https://www.ncbi.nlm.nih.gov/data-hub/genome/GCA_900231445.1/, accessed on 13 January 2023) based on the positions of SSR markers in the durum wheat reference sequence (https://wheat.pw.usda.gov/GG3/jbrowse_Durum_Svevo, accessed on 13 January 2023). Identification of upregulated gene models under abiotic stress at the seedling stage was carried out using the RNAseq data at http://www.wheat-expression.com/ (accessed on 13 January 2023), using gene models from the NCBI database.

### 4.5. Statistical Analysis

The statistical analyses of phenotypic data were performed using R Studio (Version R-4.1.0, R Foundation for Statistical Computing, Vienna, Austria). Two-way analysis of variance (ANOVA) was conducted at a significance level of 5% using the aov() function. At the same time, one-way ANOVA was used to test the variance component of each trait under each treatment, with genotype as a factor. Fisher’s least significant difference (LSD) test was used for means comparisons. Pearson correlation coefficients were calculated using the corrplot function R package [80]. Principal component analysis was performed using the prcomp() function, and then a biplot was generated with the ggbiplot function R package [80]. The genetic distance [36] matrix was utilized to construct, using GDA software [81]. The Mantel test to compare genetic distance and Euclidean matrices was computed by GenAlEx, an Excel macro realized by Peakall and Smouse [82].

## 5. Conclusions

The identification of molecular markers linked to root system traits is very important for marker-assisted selection in breeding programs. The SSR markers linked to genetic regions associated with root traits in this study were effective in discriminating the genotypes and can be used to screen candidate lines for salt tolerance. Nevertheless, the narrow root angle is not a good indicator of salt tolerance since there are more root system adaptation strategies that depend on the inner capacity of genotypes to maintain an efficient amount of roots. Moreover, present results of phenotypic response to salt stress using Win-RHIZO Pro software together with molecular analysis can provide useful highlights to increase the efficiency of plant breeding programs.

## Figures and Tables

**Figure 1 plants-12-00412-f001:**
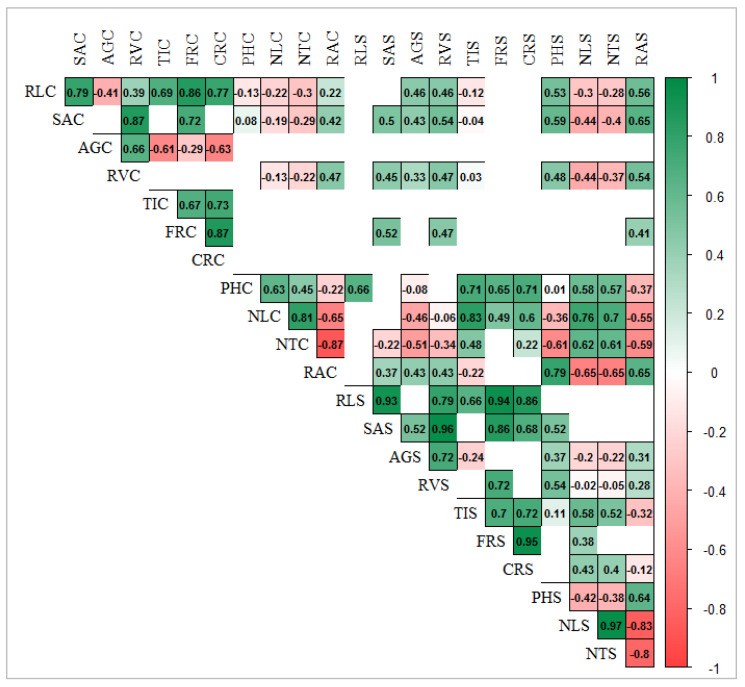
Correlation matrix among all the recorded traits in pots experiment under C-control and S-salt conditions. Positive (in green) and negative (in red) indicate significant (*p* < 0.05) correlations. For abbreviations, please see the abbreviation list at the end of the paper.

**Figure 2 plants-12-00412-f002:**
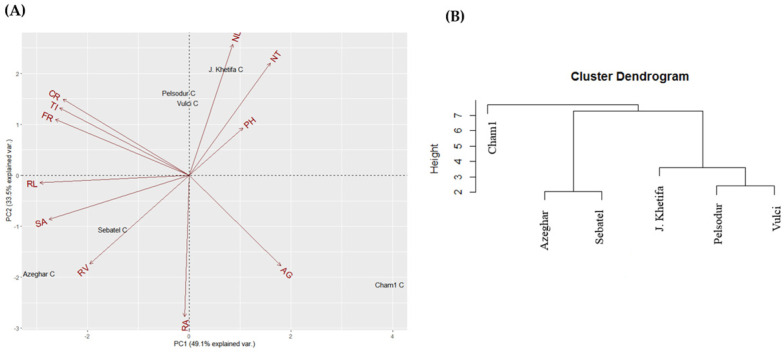
Biplot of the principal component analysis (**A**) and hierarchical analysis (**B**) of the shoot and root traits under control conditions. For abbreviations, please see the abbreviation list at the end of the paper.

**Figure 3 plants-12-00412-f003:**
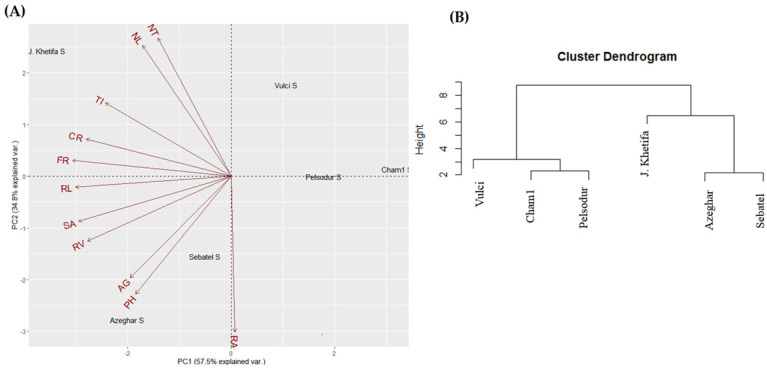
Biplot of the principal component analysis (**A**) and hierarchical analysis (**B**) of the shoot and root traits under salt conditions. For abbreviations, please see the abbreviation list at the end of the paper.

**Figure 4 plants-12-00412-f004:**
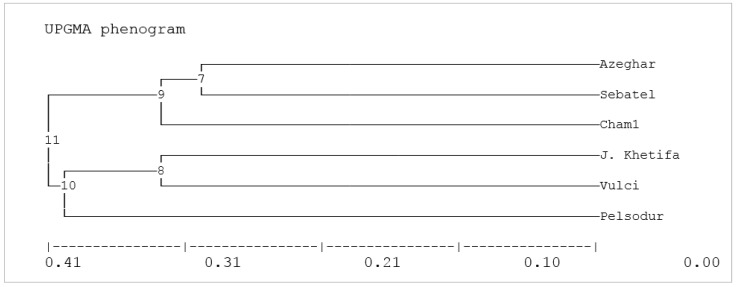
UPGMA phenogram of the Nei genetic distance among six durum wheat genotypes based on 11 SSRs markers.

**Figure 5 plants-12-00412-f005:**
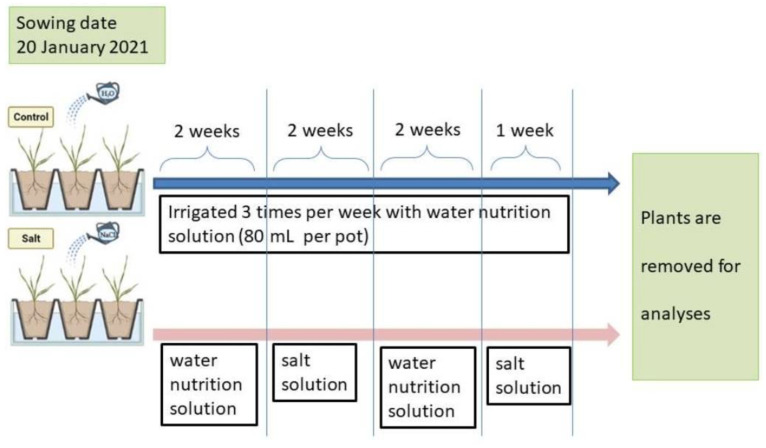
Experimental model, indicating when salt stress was applied by adding NaCl to the water nutrition solution.

**Figure 6 plants-12-00412-f006:**
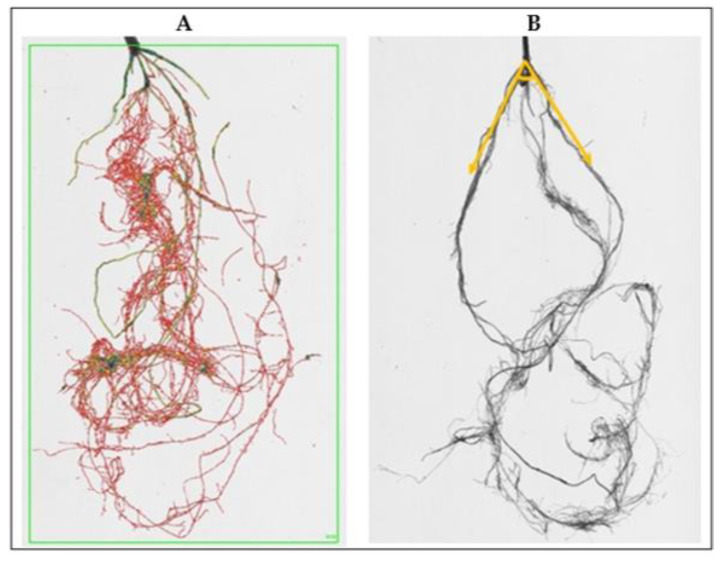
The measurements of morphological root traits (**A**) using Win-RHIZO system and root angle (**B**) using ImageJ (°).

**Table 1 plants-12-00412-t001:** Salt versus control effect on shoot traits of six durum wheat genotypes.

Genotype	Seedling Length, cm			Number of Leaves			Number of Tillers		
	Control	Salt	Diff. (%)	Control	Salt	Diff. (%)	Control	Salt	Diff. (%)
Azeghar	34.6 ± 3.2 b	40.5 ± 1.6 a	16.96 ns	5.0 ± 0.0 c	6.7 ± 0.6 c	33.3 ns	1.0 ± 0.0 c	1.0 ± 0.0 b	0.0 ns
Cham1	37.1 ± 0.6 ab	31.3 ± 0.8 bc	−15.71 ns	6.0 ± 1.7 c	6.3 ± 1.2 c	5.6 ns	2.3 ± 0.6 b	1.0 ± 0.0 b	−57.1 **
J. Khetifa	42.8 ± 1.1 a	33.1 ± 2.6 b	−22.74 *	9.3 ± 0.6 a	14.0 ±0.0 a	50.0 ***	3.0 ± 0.0 ab	3.3 ± 0.6 a	11.1 ns
Pelsodur	34.0 ± 1.5 b	29.2 ± 2.2 c	−14.18 ns	8.0 ± 0.0 ab	7.3 ± 1.5 c	−8.3 ns	3.3 ± 0.6 a	1.3 ± 0.6 b	−60.0 ***
Sebatel	36.4 ± 2.8 b	34.8 ± 0.8 b	−4.40 ns	5.7 ± 0.6 c	7.3 ± 0.6 c	29.4 ns	1.0 ± 0.0 c	1.0 ± 0.0 b	0.0 ns
Vulci	35.2 ± 7.0 b	27.6 ± 2.4 c	−21.76 ns	6.7 ± 1.5 bc	11.3 ± 1.2 b	70.0 ***	2.3 ± 0.6 b	2.7 ± 0.6 a	14.3 ns
Genotype	**			***			***		
Treatment	***			***			**		
G × T	**			***			***		

Values are means ± standard deviations (*n* = 3). Means with the same letter in the column of each treatment are not significantly different between genotypes (*p* < 0.05) (LSD test). Diff.% reports the differences between Control (column) and Salt (column). It is computed as [Diff.% = ((salt/control) − 1) × 100], and its statistic is indicated as ns—Not significant; *, **, and *** indicate significance at *p* < 0.05, *p* < 0.01, and *p* < 0.001 levels, respectively.

**Table 2 plants-12-00412-t002:** Analysis of variance (ANOVA) for root traits.

**ANOVA**	**Root Volume**	**Root Length**	**Tips**	**Root Surface Area**
Genotype	*	*	*	**
Treatments	***	***	***	***
G × T	ns	ns	*	ns
**ANOVA**	**Root Angle**	**Crossings**	**Forks**	**Average Root Diameter**
Genotype	***	*	**	ns
Treatments	***	*	**	ns
G × T	***	ns	ns	ns

A description is added, i.e., ns—Not significant; *, **, and *** indicate significance at *p* < 0.05, *p* < 0.01, and *p* < 0.001 levels, respectively.

**Table 3 plants-12-00412-t003:** Salt versus control effect on root traits of six durum wheat genotypes.

Trait	Treat	Genotype					
		Azeghar	Cham1	J. Khetifa	Pelsodur	Sebatel	Vulci
**Root volume (cm^3^)**	control	0.80 ± 0.28 a	0.57 ± 0.15 a	0.57 ± 0.04 a	0.61 ± 0.19 a	0.68 ± 0.18 a	0.48 ± 0.18 a
	salt	1.28 ± 0.24 a	0.57 ± 0.02 c	1.07 ± 0.34 ab	0.68 ± 0.06 bc	0.99 ± 0.15 abc	0.77 ± 0.29 bc
**Diff. (%)**		**59.6 ***	**0.0 ns**	**88.8 ****	**12.4 ns**	**45.1 ns**	**61.5 ns**
**Root Length (cm)**	control	684 ± 106 a	333 ± 73 c	475 ± 104 bc	521 ± 80 abc	578 ± 55 ab	530 ± 120 ab
	salt	826 ± 52 ab	603 ± 23 b	890 ± 192 a	590 ± 7 b	737 ± 200 ab	627 ± 155 b
**Diff. (%)**		**20.6 ns**	**81.2 ***	**87.3 ****	**13.4 ns**	**27.6 ns**	**18.2 ns**
**Number of tips**	control	1243 ± 287 a	633 ± 94 a	1188 ± 406 a	1214 ± 225 a	1326 ± 507 a	1272 ± 353 a
	salt	1510 ± 119 b	1362 ± 21 b	2399 ± 143 a	1401 ± 265 b	1480 ± 352 b	1311 ± 309 b
**Diff. (%)**		**21.5 ns**	**115.3 ***	**101.9 *****	**15.4 ns**	**11.6 ns**	**3.0 ns**
**Root surface area (cm^2^)**	control	81.4 ± 7.8 a	48.5 ± 11.8 b	57.7 ± 4.2 b	62.8 ± 14.5 ab	69.9 ± 11.6 ab	56.4 ± 16.5 b
	salt	114.9 ± 14 a	65.5 ± 0.0 c	109 ± 28 ab	71.1 ± 2.6 bc	95.3 ± 20.3 abc	77.2 ± 21.7 bc
**Diff. (%)**		**41.2 ***	**35.1 ns**	**89.0 ****	**13.2 ns**	**36.3 ns**	**37.0 ns**
**Root angle (°)**	control	110.1 ± 4.4 a	109.4 ± 7.4 a	90.1 ± 3.3 b	79.7 ± 1.3 c	106.1 ± 2.3 a	84.9 ± 2.8 bc
	salt	117.0 ± 7.9 a	104.5 ± 3.2 bc	95.4 ± 4.3 d	109.5 ± 6.6 ab	115.3 ± 2.7 a	96.7 ± 2.9 cd
**Diff. (%)**		**6.2 ns**	**−4.5 ns**	**6.0 ns**	**37.3 *****	**8.7 ***	**13.9 ***
**Crossings**	control	956 ± 487 a	421 ± 162 a	942 ± 204 a	887 ± 62 a	857 ± 136 a	859 ± 218 a
	salt	1056 ± 97 abc	512 ± 315 c	1543 ± 423 a	981 ± 241 abc	1122 ± 503 ab	916 ± 266 bc
**Diff. (%)**		**10.5 ns**	**21.8 ns**	**63.9 ***	**10.6 ns**	**30.9 ns**	**6.6 ns**
**Forks**	control	4990 ± 914 a	2774 ± 542 b	4651 ± 511 a	4736 ± 531 a	4587 ± 897 a	4152 ± 931 ab
	salt	6606 ± 143 ab	3547 ± 242 c	8180 ± 1987 a	4801 ± 183 bc	6100 ± 2120 abc	5111 ± 1201 bc
		**32.4 ns**	**27.9 ns**	**75.9 ***	**1.4 ns**	**33.0 ns**	**23.1 ns**
**Mean diameter (mm)**	control	0.39 ± 0.1 ab	0.46 ± 0.01 a	0.39 ± 0.06 ab	0.38 ± 0.03 ab	0.38 ± 0.05 ab	0.33 ± 0.3 b
	salt	0.44 ± 0.03 a	0.35 ± 0.01 b	0.39 ± 0.03 ab	0.39 ± 0.02 ab	0.42 ± 0.03 ab	0.39 ± 0.06 ab
**Diff. (%)**		**12.8 ns**	**−23.9 ***	**−1.7 ns**	**1.3 ns**	**8.7 ns**	**17.0 ns**

Values are means ± standard deviations (*n* = 3). Means with the same letter in each row are not significantly different between genotypes (*p* < 0.05) (LSD test). The differences between Control and Salt [Diff.% = ((salt/control) − 1)*100] and its statistic is indicated as ns—Not significant; *, **, and *** indicate significance at *p* < 0.05, *p* < 0.01, and *p* < 0.001 levels, respectively.

**Table 4 plants-12-00412-t004:** Genetic diversity in durum wheat based on 11 SSR markers identifying 13 loci.

Locus	Na	I	Ho	He	uHe	PIC	Size, bp
cfa2086	5	1.561	0.000	0.778	0.848	0.744	223–299
wms5	3	1.011	0.000	0.611	0.667	0.536	180–190
gwm234a	2	0.451	0.000	0.278	0.303	0.240	214–218
gwm234b	2	0.693	0.000	0.500	0.545	0.375	242–256
wmc727a	2	0.679	0.833	0.486	0.530	0.368	88–96
wmc727b	2	0.451	0.000	0.278	0.303	0.240	null-231
cfa2257	5	1.424	0.333	0.722	0.788	0.680	104–156
wms205	2	0.693	1.000	0.500	0.545	0.375	157–170
gwm427	4	1.330	0.000	0.722	0.788	0.672	201–256
gwm573.2	5	1.424	1.000	0.722	0.788	0.680	181–240
gwm636	3	1.011	0.000	0.611	0.667	0.536	110–131
gwm459a	3	1.011	0.000	0.611	0.667	0.536	131–153
gwm499b	4	1.330	0.000	0.722	0.788	0.672	147–197
mean	3.2	1.005	0.244	0.580	0.633		

Na—No. of Alleles, I—Shannon’s Information Index = −1 * Sum (pi * Ln (pi)), Ho—Observed Heterozygosity, He—Expected Heterozygosity, uHe—Unbiased Expected Heterozygosity, PIC—polymorphism information content. Different locus of same marker are highlighted as “a” and “b”.

**Table 5 plants-12-00412-t005:** Private alleles (defined by a set of populations currently active).

Locus	Allele	Frequency	Found in
cfa2086	223	1	Sebatel
cfa2086	299	1	Pelsodur
cfa2086	269	1	Cham
cfa2086	277	1	J. Khetifa
wms5	185	1	Cham
gwm234a	214	1	J. Khetifa
wmc727a	1	1	Pelsodur
cfa2257	121	1	Cham
cfa2257	142	0.5	Azeghar
cfa2257	104	0.5	Azeghar
gwm427	237	1	Sebatel
gwm427	201	1	Azeghar
gwm573.2	226	0.5	Pelsodur
gwm573.2	181	0.5	Azeghar
gwm636	131	1	Pelsodur
gwm459a	131	1	Vulci
gwm499b	147	1	Sebatel
gwm499b	154	1	J. Khetifa

Different locus of same marker are highlighted as “a” and “b”.

**Table 6 plants-12-00412-t006:** Primers used for PCR amplification.

Marker	Primer Forward (5′-3′)	Primer Reverse (5′-3′)	Chr.	AT (°C)
cfa2086	TCTACTTTCAGGGCACCTCG	TCTCTCCAAACCTCCCTGTAA	2A	56
gwm573.2	AAGAGATAACATGCAAGAAA	TTCAAATATGTGGGAACTAC	7B	45
wmc727	CATAATCAGGACAGCCGCAC	TAGTGGCCTGATGTATCTAGTTGG	5A	55
wms205	CGACCCGGTTCACTTCAG	AGTCGCCGTTGTATAGTGCC	5A	56
wms5	GCCAGCTACCTCGATACAACTC	AGAAAGGGCCAGGCTAGTAGT	3A	56
gmw459	ATGGAGTGGTCACACTTTGAA	AGCTTCTCTGACCAACTTCTCG	6A	54
gwm234	GAGTCCTGATGTGAAGCTGTTG	CTCATTGGGGTGTGTACGTG	5B	55
gwm427	AAACTTAGAACTGTAATTTCAGA	AGTGTGTTCATTTGACAGTT	6A	45
gwm499	ACTTGTATGCTCCATTGATTGG	GGGAGTGGAAACTGCATAA	5B	52
gwm637	AAAGAGGTCTGCCGCTAACA	TATACGGTTTTGTGAGGGGG	4A	55
cfa2257	GATACAATAGGTGCCTCCGC	CCATTATGTAAATGCTTCTGTTTGA	7A	49

## Data Availability

https://doi.org/10.5281/zenodo.7534377 (accessed on 20 December 2022).

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
