# Peer review of "Phenotypic and Genotypic Diversity of Roots Response to Salt in Durum Wheat Seedlings"

_plants, 2023, doi:10.3390/plants12020412_

Round 1
Reviewer 1 Report
My comments are as under:
It is required that the most significant findings of the author be included in the abstract.
In the introduction, you should state your hypothesis explicitly and then set the stage for the following paragraph.
There is a lot of unnecessary repetition throughout the whole thing.
A careless approach was taken in the writing of the figure ligands.
There is a possibility that this conversation will include other significant information and sources in the future.
Rearrange the information, and give the concluding section a thorough reading through while also editing it.
Author Response
Answers to REVIEWER 1
My comments are as under:
It is required that the most significant findings of the author be included in the abstract.
Thanks for the suggestion, the abstract was modified accordingly.
In the introduction, you should state your hypothesis explicitly and then set the stage for the following paragraph.
Thanks for the suggestion, the last paragraph of the introduction was integrated accordingly.
There is a lot of unnecessary repetition throughout the whole thing.
We are sorry, but we are not sure to fully understand what reviewer 1 refers to. In any event, we edit the entire document and modified the discussion to reduce fragmentation inserting some numbering lists to make it easier to follow the discussion.
A careless approach was taken in the writing of the figure ligands.
Thanks for the suggestion, Legend of figures 1-4 were revised and integrated.
There is a possibility that this conversation will include other significant information and sources in the future.
We are sorry, but we are not sure to fully understand reviewer 1 point and to which conversation reviewer 1 refers. Regarding future utilization of present results, it is stated in the conclusion section which was modified to make it clearer “Moreover, present results of phenotypic response to salt stress using Win-RHIZO Pro software together with molecular analysis can provide useful highlights to increase the efficiency of plant breeding programs.”
Rearrange the information, and give the concluding section a thorough reading through while also editing it.
The conclusion section is slightly changed, please see the previous answer.

Reviewer 2 Report
The article by Urbanavičiūtė et al describes the detailed analysis of phenotypes associated with salt-tolerance in six wheat varieties and SSR markers associated with root traits. It is shown that salt-induced changes in plant morphology vary in different genotypes, in shoots and roots. The suggestion of a relationship between salt tolerance and root angle has not been confirmed for all genotypes.
The study focuses on morphological characteristics, while physiological and biochemical characteristics remain outside the scope of the research. The article contains valuable information for breeders, however, the presentation of the data obtained requires improvement.
The material presented is rather poorly illustrated. This can be partially corrected by redistributing the illustrative data between the main part of the article and the supplementary materials. Also, the data on PCR amplifications of SSR markers are not shown. At least a representative gel showing the differences in molecular weights of SSR markers should be presented.
One of the main studied indicators is the root angle. In this study the plants were grown in the sand, and the angle was determined after the roots are excavated from the sand. It should be confirmed that this approach allows the correct determination of the root angle in plants growing under control and stress conditions (or the reference to other published works confirming this should be presented).
The discussion includes the description of the candidate genes, but this description is very fragmentary and incoherent. For example, it is not clear what the sentence "Abscisic acid is a substrate of the ABC transporter that is involved in the active transport of various molecules across biological membranes, such as heavy metals, lipids, glucosinolates, and phytohormones” means. The possible connection of candidate genes with stress tolerance should be discussed.
Minor comments:
Table 1, it should be explained what exactly the “Diff. (%)” means, which data are compared.
Also, it is not clear why seedling length is designated as “PH” (example, line 99) and the number of tips – as “TI” (line 116).
The use of the term “overexpressed” is not correct in several places of the text. Examples: “All genes were overexpressed in seedlings root under…” – line 262; “overexpressed in seedlings root under abiotic stress…” – line 268. The term “up-regulated” (lines 292-293) is more appropriate.
Line 355 “thioredoxin … is involved in oxidative deactivation of photosynthesis…” Should it be “deactivation after/from oxidative damage?
Please, check the phrase (line 371) “some SSR markers' rare alleles could help to discriminate salt-tolerant candidates in the next generations”. Discriminate or select?
Table S3 – gene starting and ending positions are not filled correctly for marker gwm573.2
Author Response
Answers to REVIEWER 2
The article by Urbanavičiūtė et al describes the detailed analysis of phenotypes associated with salt-tolerance in six wheat varieties and SSR markers associated with root traits. It is shown that salt-induced changes in plant morphology vary in different genotypes, in shoots and roots. The suggestion of a relationship between salt tolerance and root angle has not been confirmed for all genotypes.
The study focuses on morphological characteristics, while physiological and biochemical characteristics remain outside the scope of the research. The article contains valuable information for breeders, however, the presentation of the data obtained requires improvement.
The material presented is rather poorly illustrated. This can be partially corrected by redistributing the illustrative data between the main part of the article and the supplementary materials. Also, the data on PCR amplifications of SSR markers are not shown. At least a representative gel showing the differences in molecular weights of SSR markers should be presented.
Thanks for the suggestion. The SSR Excel output was added in the supplementary material and a photo of the pot experiment as Fig 5 in the main test and one as Fig S1 in the supplementary.
One of the main studied indicators is the root angle. In this study the plants were grown in the sand, and the angle was determined after the roots are excavated from the sand. It should be confirmed that this approach allows the correct determination of the root angle in plants growing under control and stress conditions (or the reference to other published works confirming this should be presented).
Thanks for the comments which are important point. The point could be answered by looking at several aspects.
1) Yes, several other studies were run on root under sand; some of them are reported in our manuscript as:
[23] El Hassouni et al. (2018) test Root System Traits Using the Pasta Strainer in sand soil.
[24] Wasson et al. (2012) studied the root system in sandy soil
[25] Urbanaviči et al. (2022) studied root reactions in sand.
Most of the cited papers (and references therein) were detecting root conformation (including root depth and root angle) under drought stress.
2) To answer if root the angle utilizing sand is comparable with the one in field soil (which is not our aim) the best studies are the ones utilizing sand under different pressure and mechanical impedance. Their results are the following:
- Collis-George N. & Yoganathan P. (1985a) [The effect of soil strength on germination and emergence of wheat (Triticum aestivum L.) I. Low shear strength conditions. Australian Journal of Soil Research 23, 577– 587.] They determined that different compression in soil (sand with externally applied loads) does not affect the germination rate nor the root angle, but the root diameters.
- While Whalley WR, Finch-Savage WE, Cope RE, Rowse HR, Bird NRA (1999) [The response of carrot (Daucus carota L.) and onion (Allium cepa L.) seedlings to mechanical impedance and water stress at sub-optimal temperatures. Plant Cell Environ 22:229–242. doi:10.1046/j.1365-3040.1999.00412.x] revealed, in different species, that the mechanical impedance has a stronger effect on the shoot than on the root.
- Jin, K., Shen, J., Ashton, R.W. et al. 2015 [The effect of impedance to root growth on plant architecture in wheat. Plant Soil 392, 323–332. https://doi.org/10.1007/s11104-015-2462-0] detected a significant impedance effect in decreasing root growth angle from 55° to 43° (from the vertical), with impedance causing steeper roots.
3) The main point is not if sand root angle is different than field soil root angle, but i) if salt affects root angle versus control conditions and ii) to have an easy and precise method to record root angle (as the other root traits). The sand was used to have an easier way to extract the root from the media. Even if sand would affect root angle compared with other media, with sand we can have precise measurements, while growing plants in field soil this is not possible. In any event, the comparison was between salt and control both under sand.
The discussion includes the description of the candidate genes, but this description is very fragmentary and incoherent. For example, it is not clear what the sentence "Abscisic acid is a substrate of the ABC transporter that is involved in the active transport of various molecules across biological membranes, such as heavy metals, lipids, glucosinolates, and phytohormones” means. The possible connection of candidate genes with stress tolerance should be discussed.
Thanks for the comments. We reduced fragmentation by inserting also some numbered lists that would allow us to better follow the discussion. We also modified the text to explain better the sentence regarding ABC.
Minor comments:
Table 1, it should be explained what exactly the “Diff. (%)” means, which data are compared.
Thanks for the comment. The description was in the Table footnote, me modify it to make it clearer.
Also, it is not clear why seedling length is designated as “PH” (example, line 99) and the number of tips – as “TI” (line 116).
These are simply abbreviations as stated in the “Abbreviation list” the choice was PH for plant height (but plant height is usually considered in the adult phase) TI is for Tips.
The use of the term “overexpressed” is not correct in several places of the text. Examples: “All genes were overexpressed in seedlings root under…” – line 262; “overexpressed in seedlings root under abiotic stress…” – line 268. The term “up-regulated” (lines 292-293) is more appropriate.
Thanks for the suggestion we substituted overexpressed with up-regulated.
Line 355 “thioredoxin … is involved in oxidative deactivation of photosynthesis…” Should it be “deactivation after/from oxidative damage?
We are not sure to fully understand the reviewer 2’s point. The oxidative damage deactivates photosynthesis and the thioredoxin protects from that. So “after”. The sentence was changed into: “…. protects cells from oxidative damage, and protects plants from oxidative deactivation of photosynthesis-related enzymes”.
Please, check the phrase (line 371) “some SSR markers' rare alleles could help to discriminate salt-tolerant candidates in the next generations”. Discriminate or select?
Discriminate was chosen since it allows to “distinguish” among genotypes which is the condition to then select. But we agree with reviewer select is more direct, so the word was changed as suggested.
Table S3 – gene starting and ending positions are not filled correctly for marker gwm573.2
Thanks for the indication, we correct gwm573.2 position in Table S3.

Round 2
Reviewer 2 Report
I am completely satisfied with the detailed answers, and I think the article fully meets the requirements of the journal.
Minor comment: Line 434 – I still suggest using “REactivation after oxidative damage”, but I leave the choice of wording to the authors.